# The Effect of Aging on Nitric Oxide Production during Cerebral Ischemia and Reperfusion in Wistar Rats and Spontaneous Hypertensive Rats: An In Vivo Microdialysis Study

**DOI:** 10.3390/ijms241612749

**Published:** 2023-08-13

**Authors:** Yasuo Ito, Harumitsu Nagoya, Masamizu Yamazato, Yoshio Asano, Masahiko Sawada, Tomokazu Shimazu, Makiko Hirayama, Toshimasa Yamamoto, Nobuo Araki

**Affiliations:** Department of Neurology, Saitama Medical University, 38 Morohongo, Moroyama, Saitama 350-0495, Japanarakin@saitama-med.ac.jp (N.A.)

**Keywords:** nitric oxide (NO), in vivo microdialysis, forebrain ischemia, striatum, hippocampus

## Abstract

Nitric oxide (NO) is involved in the pathogenesis of cerebral ischemic injury. Here, we investigated the effects of aging on NO production during cerebral ischemia-reperfusion (IR). Male Wister rats (WRs) were assigned to 12-month-old (older; *n* = 5) and 3-month-old (younger; *n* = 7) groups. Similarly, male spontaneous hypertensive rats (SHRs) were allocated to 12-month-old (older; *n* = 6) and 3-month-old (younger; *n* = 8) groups. After anesthesia, their NO production was monitored using in vivo microdialysis probes inserted into the left striatum and hippocampus. Forebrain cerebral IR injuries were produced via ligation of the bilateral common carotid arteries, followed by reperfusion. The change in the NO_3_^−^ of the older rats in the SHR groups in the striatum was less compared to that of the younger rats before ischemia, during ischemia, and after reperfusion (*p* < 0.05). In the hippocampus, the change in the NO_3_^−^ of the older rats in the SHR groups was lower compared to that of the younger rats after reperfusion (*p* < 0.05). There were no significant differences between the two WR groups. Our findings suggested that aging in SHRs affected NO production, especially in the striatum, before and during cerebral ischemia, and after reperfusion. Hypertension and aging may be important factors impacting NO production in brain IR injury.

## 1. Introduction

Nitric oxide (NO) is an important molecule in the central nervous system [1] and is associated with the pathogenesis of cerebral ischemic injury. During acute cerebral ischemia, NO production and NO synthase (NOS) activity are elevated, and have been suggested to be cytotoxic to the brain [2].

In a mouse study, we previously investigated the kinetics of NO production by endothelial NOS (eNOS) and neuronal NOS (nNOS) after transient global forebrain ischemia, and furthermore, we investigated the NO production and ischemic changes in the striatum in eNOS knockout and nNOS knockout mice during cerebral IR. The in vivo findings suggested that the NO production in the striatum after reperfusion was closely related to the activities of both nNOS and eNOS, and was mainly related to nNOS following reperfusion [3]. More recently, we investigated the effects of edaravone on NO production, hydroxyl radical (OH^−^) metabolism, and nNOS expression during cerebral IR injury. The in vivo findings suggested that edaravone exerted a neuroprotective effect by reducing the levels of OH^−^ metabolites, increasing the NO production, and decreasing the nNOS expression in brain cells [4]. Hypertension and aging are major risk factors for arteriosclerosis and stroke. However, the effects of hypertension and aging on NO production during cerebral ischemia are unclear. Compared to WRs with normal blood pressure, SHRs are used as a model of hypertension [5]. It is known that cerebral infarction is worse in SHRs than WRs, as shown in studies using middle cerebral artery occlusion [5], and the concentration of NO release, detected using electrochemical microsensors, is significantly lower in spontaneously hypertensive rats (SHR-SP) than that in Spraigue-Dawley (SD) rats [6].

In a study on the expression of the NOS mRNA and protein levels in cerebrovascular SHRs, both the eNOS mRNA and eNOS protein levels were lower than those in WRs, but both the inducible NOS (iNOS) mRNA and iNOS protein levels were high [7]. 

In a study on the NOS expression in the brainstem of 12-week-old SHRs, the nNOS and iNOS expressions were reduced and the total NOS activity was also lower than that in WRs. This leads to the downregulation of nNOS and iNOS in SHRs [8]. Thus, the relationship between NOS expression, NO activity, and hypertension remains unknown.

Aging is more closely associated with a marked impairment in the determinants of NO generation [9] and a decline in physiological function, such as NO bioavailability [10]

In the present study, we investigated the effects of aging and hypertension on NO production during cerebral IR in WRs and SHRs.

## 2. Results

### 2.1. MABP

The pre-ischemic MABP baselines are shown as averages of the last 40 min before ischemia (Figure 1).

#### 2.1.1. WRs

In the WR group, the MABPs were significantly higher in the older rats (145 ± 28 mmHg) than in the younger rats (111 ± 27 mmHg) before ischemia and 10–40 min after reperfusion (*p* < 0.05) (Figure 1A).

#### 2.1.2. SHRs

In the SHR group, the MABPs were significantly higher the in older rats (166 ± 20 mmHg) than in the younger rats (118 ± 17 mmHg) 10 min after reperfusion (*p* < 0.05) (Figure 1B).

#### 2.1.3. Comparison of MABPs between WRs and SHRs

In the older rats, the MABPs were significantly higher in the SHRs (166 ± 25 mmHg) than in the WRs (129 ± 13 mmHg) 40 min before ischemia and 10 min after reperfusion (*p* < 0.05).

In the younger rats, the MABPs were significantly higher in the SHRs (127 ± 13 mmHg) than in the WRs (104 ± 14 mmHg) 40–10 min before ischemia and 10–40 min after reperfusion (*p* < 0.05)

### 2.2. rCBF

#### 2.2.1. WRs

In the striatums of the WR group, the rCBF was significantly lower in the older rats (16.4 ± 10.4% of baseline) compared to the younger rats (30.2 ± 7.5%) during ischemia (*p* < 0.05) (Figure 2A). In contrast, the rCBF was significantly higher in the older rats (179.3 ± 35.8% of baseline) than in the younger rats (130.7 ± 31.3%) 20–40 min after reperfusion (*p* < 0.05) (Figure 2A). In the WR group, the hippocampal rCBF was significantly higher in the older rats (166.9 ± 35.7% of baseline) than in the younger rats (119.5 ± 16.3%) 20 min after reperfusion (*p* < 0.05) (Figure 2B).

#### 2.2.2. SHRs

In the striatums of the SHR group, the rCBF values were similar between the two age groups (Figure 2C). In contrast, in the hippocampi of the SHR group, the rCBF was significantly lower in the older rats (17.9 ± 6.3% of baseline) compared to the younger rats (35.8 ± 12.4%) during ischemia (*p* < 0.05), and significantly higher in the older animals (213.6 ± 48.7% of baseline) than in the younger animals (163.8 ± 33.6%) 10 min after reperfusion (*p* < 0.05) (Figure 2D).

### 2.3. NO Metabolites

#### 2.3.1. In the Striatum in WRs (Figure 3)

##### NO_2_^−^

The NO_2_^−^ levels were significantly higher in the older rats (1.34 ± 0.52 µmol/L) than in the younger rats (0.81 ± 0.19 µmol/L) during ischemia, but lower in the former group (1.00 ± 0.36 µmol/L) than in the latter group (1.79 ± 0.68 µmol/L) 10 min after the start of reperfusion (*p* < 0.05) (Figure 3A).

**Figure 3 ijms-24-12749-f003:**
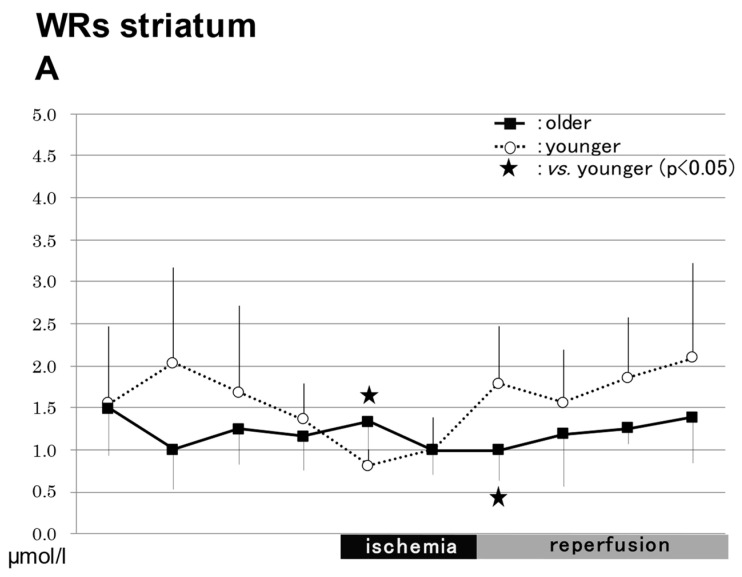
NO production in WRs. Changes in nitric oxide production in WRs are shown in (**A**–**F**). The NO_2_^−^ levels were significantly higher in the older rats (1.34 ± 0.52 μmol/L) than in the younger rats (0.81 ± 0.19 μmol/L) during ischemia, but lower in the former group (1.00 ± 0.36 μmol/L) than in the latter group (1.79 ± 0.68 μmol/L) 10 min after the start of reperfusion (*p* < 0.05) (**A**). The levels of NO_3_^−^ did not differ significantly between the older and younger rats before or during ischemia, or after reperfusion (**B**). The levels of total NO also did not differ significantly between the older and younger rats before or during ischemia, or after reperfusion (**C**). The levels of NO_2_^−^ did not differ significantly between the older and younger rats before or during ischemia, or after reperfusion (**D**). The levels of NO_3_^−^ did not differ significantly between the older and younger rats before or during ischemia, or after reperfusion (**E**). The levels of total NO also did not differ significantly between the older and younger rats before or during ischemia, or after reperfusion (**F**). WRs, Wistar rats; NO, nitric oxide.

##### NO_3_^−^

The levels of NO_3_^−^ did not differ significantly between the older and younger rats before or during ischemia, or after reperfusion (Figure 3B).

##### Total NO

The levels of total NO also did not differ significantly between the older and younger rats before or during ischemia, or after reperfusion (Figure 3C).

#### 2.3.2. In the Hippocampus in WRs (Figure 3)

##### NO_2_^−^

The levels of NO_2_^−^ did not differ significantly between the older and younger rats before or during ischemia, or after reperfusion (Figure 3D).

##### NO_3_^−^

The levels of NO_3_^−^ did not differ significantly between the older and younger rats before or during ischemia, or after reperfusion (Figure 3E).

##### Total NO

The levels of total NO also did not differ significantly between the older and younger rats before or during ischemia, or after reperfusion (Figure 3F).

#### 2.3.3. In the Striatum in SHRs (Figure 4)

##### NO_2_^−^

The levels of NO_2_^−^ did not differ significantly between the older and younger rats before or during ischemia, or after reperfusion (Figure 4A).

**Figure 4 ijms-24-12749-f004:**
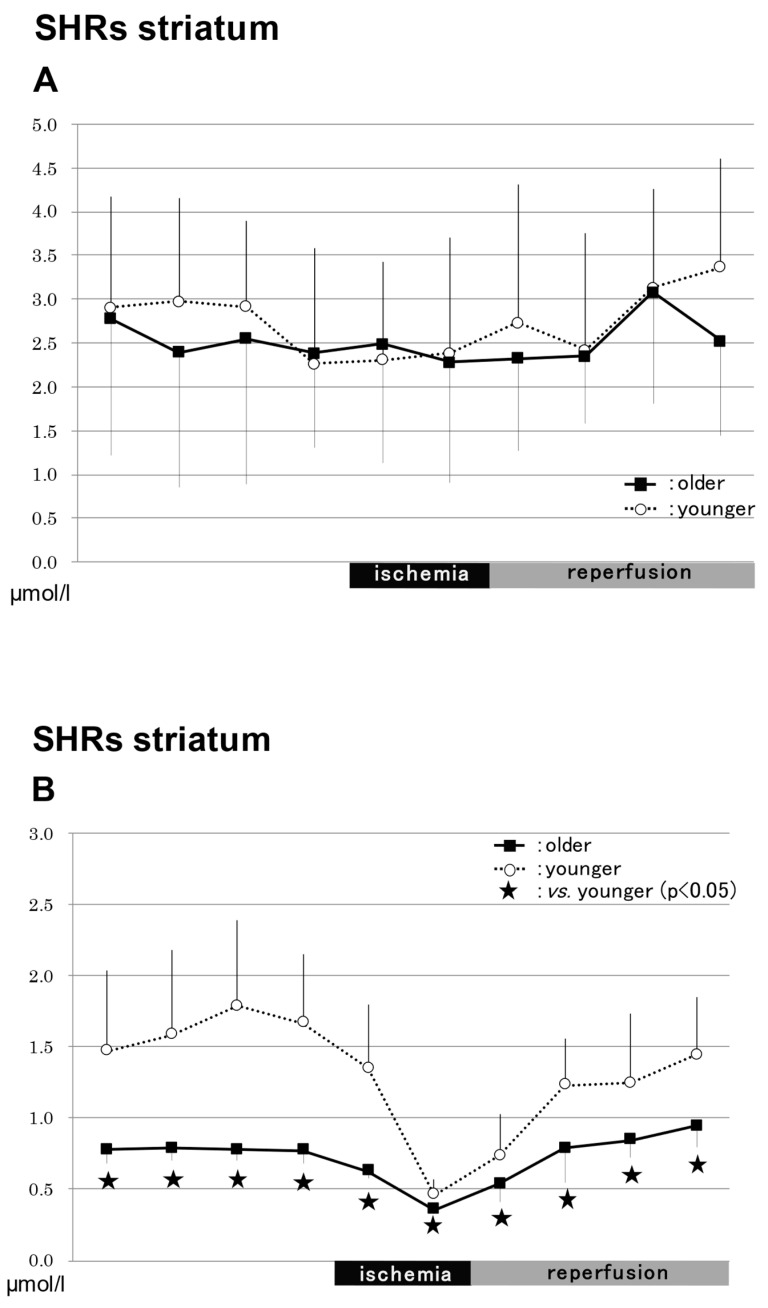
NO production in SHRs. Changes in NO production in SHRs are shown in (**A**–**F**). The levels of NO_2_^−^ did not differ significantly between the older and younger rats before or during ischemia, or after reperfusion (**A**). The NO_3_^−^ levels were significantly lower in the older rats (0.78 ± 0.10 μmol/L) compared to the younger rats (1.47 ± 0.56 μmol/L) before and after ischemia, and after the start of reperfusion (*p* < 0.05) (**B**). The total NO levels were significantly lower in the older rats (3.5 ± 1.0 μmol/L) compared to the younger rats (4.8 ± 1.13 μmol/L) 40 min after the start of reperfusion (*p* < 0.05) (**C**). The levels of NO_2_^−^ did not differ significantly between the older and younger rats before or during ischemia, or after reperfusion (**D**). The NO_3_^−^ levels were significantly lower in the older rats (0.78 ± 0.20 μmol/L) than in the younger rats (1.67 ± 0.70 μmol/L) 20–40 min after the start of reperfusion (*p* < 0.05) (**E**). The total NO levels were significantly lower in the older rats (3.4 ± 1.2 μmol /L) compared to the younger rats (5.2 ± 2.5 μmol/L) 30 min before ischemia and 30–40 min after the start of reperfusion (*p* < 0.05) (**F**). SHRs, spontaneously hypertensive rats; NO, nitric oxide.

##### NO_3_^−^

The NO_3_^−^ levels were significantly lower in the older (0.78 ± 0.10 µmol/L) rats compared to the younger rats (1.47 ± 0.56 µmol/L) before and after ischemia, and after the start of reperfusion (*p* < 0.05) (Figure 4B).

##### Total NO

The total NO levels were significantly lower in the older rats (3.5 ± 1.0 µmol/L) compared to the younger rats (4.8 ± 1.13 µmol/L) 40 min after the start of reperfusion (*p* < 0.05) (Figure 4C).

#### 2.3.4. In the Hippocampus in SHRs (Figure 4)

##### NO_2_^−^

The levels of NO_2_^−^ did not differ significantly between the older and younger rats before or during ischemia, or after reperfusion (Figure 4D).

##### NO_3_^−^

The NO_3_^−^ levels were significantly lower in the older rats (0.78 ± 0.20 µmol/L) than in the younger rats (1.67 ± 0.70 µmol/L) 20–40 min after the start of reperfusion (*p* < 0.05) (Figure 4E).

##### Total NO

The total NO levels were significantly lower in the older rats (3.4 ± 1.2 µmol /L) compared to the younger tats (5.2 ± 2.5 µmol/L) 30 min before ischemia and 30–40 min after the start of reperfusion (*p* < 0.05) (Figure 4F).

#### 2.3.5. Comparison of Total NO between WRs and SHRs

In the striatums, the total NO in the older rats was significantly higher in the SHRs (2.8 ± 1.0 µmol/L) than that in the WRs (1.7 ± 0.2 µmol/L) only 10 min after reperfusion (*p* < 0.05). The total NO in the younger rats was significantly higher in the SHRs (2.8 ± 1.3 µmol/L) than that in the WRs (1.5 ± 0.6 µmol/L) during ischemia (*p* < 0.05).

In the hippocampi, however, the total NO was not significantly different between the WRs and SHRs.

## 3. Materials and Methods

### 3.1. Animal Groupings

The WRs and SHRs were housed in the animal care facility at Saitama Medical University.

In total, 12 male WRs were divided into 12-month-old (older; *n* = 5) and 3-month-old (younger; *n* = 7) groups. Similarly, 14 male SHRs were assigned to 12-month-old (older; *n* = 6) and 3-month-old (younger; *n* = 8) groups. The rats were anesthetized via an intraperitoneal injection of the pentobarbital sodium. Their rectal temperature was maintained at 37.0–37.5 °C with a heating pad. A polyethylene catheter (PE-50; BD, Japan) was inserted into the right femoral artery to measure the mean arterial blood pressure (MABP). A polyethylene catheter (PE-90; BD) was inserted into the right femoral vein for exsanguination. Laser Doppler probes (Advance Laser Flowmeter ALF21D; Advance, Tokyo, Japan) were inserted into the right striatum and hippocampus to continuously measure the regional cerebral blood flow (rCBF) (Figure 5). All the animal experiments were approved by the Institutional Animal Care and Use Committee of Saitama Medical University, Japan (approval number: 000317).

### 3.2. In Vivo Microdialysis

The production of NO was continuously monitored using in vivo microdialysis. A microdialysis probe was inserted into the left striatum and hippocampus (Figure 1) and perfused with Ringer’s solution at a constant rate of 2 µL/min. After a 2 h equilibration period, fractions were collected every 10 min using a micro-fraction collector (EF-80B; Eicom, Kyoto, Japan).

In both the WR and SHR groups, forebrain cerebral ischemia was produced via ligation of the bilateral common carotid arteries, and systemic hypotension (MABP < 50 mmHg) was produced via exsanguination. After 20 min, the loops around both common carotid arteries were released for reperfusion, and the exsanguinated blood was reinfused. The blood parameters were monitored for 40 min.

The nitrite (NO_2_^−^) and nitrate (NO_3_^−^) levels in the dialysates were measured using a HPLC-pump system (EP-60; Eicom, Kyoto, Japan), automatic sample injector (AS-10; Eicom), and column oven (ATC-10; Eicom) using the Griess reaction (ENO-20; Eicom), in which the NO_2_^−^ and NO_3_^−^ were separated on a packed column (NO-PAK; Eicom) and NO_3_^−^ was reduced to NO_2_^−^ in a cadmium reduction column (NO-RED; Eicom). In addition, we evaluated and compared the total NO (NO_2_^−^ + NO_3_^−^) between the old and young SHR and WR groups. For the statistical analysis, comparisons between the two groups were performed using a factorial analysis of variance (ANOVA), and the Fisher’s significance level was set to < 5%.

## 4. Discussion

In the WRs, the MABPs were higher in the older animals than the younger animals during IR. These results suggest that aging significantly increases the MABP in these animals with a normal blood pressure. In contrast, in the SHRs, no significant difference was observed before or during ischemia, or after reperfusion.

In a comparison of the MABPs between the WR group and SHR group, in the older rats, the MABPs were significantly higher in the SHRs than those in the WRs 40 min before ischemia and 10 min after reperfusion. In the younger rats, the MABPs were also significantly higher in the SHRs than those in the WRs 10–40 min before ischemia and 10–40 min after reperfusion. These results indicated that the aging in SHRs would be more accelerated than that in WRs, as the 3-month-old SHRs had already become hypertensive.

In the present study, in young SHRs aged 3 months, the MABP began to rise, and there was no significant difference between these and older animals.

When we compared the baseline MABPs between the younger SHRs and younger WRs, the blood pressure in the younger SHRs was already significantly higher than that in the younger WRs.

Katsuta et al. [11] studied the effect of aging on blood pressure in SHRs, and concluded that cerebral blood flow injury caused by hypertension can already be detected in 4-month-old rats.

A review of rCBF suggests that rCBF is more difficult to recover in older WRs than younger WRs after reperfusion. In the SHRs, although the rCBF was not significantly different between the old and young rats, both groups appeared to have a similar delay in recovery to the older WRs. Jeffrey et al. [12] investigated the changes in rCBF after cerebral ischemia and reperfusion in pigs, and showed that older animals (6–8 months of age) had a similar delay in recovery to younger animals (1–2 weeks of age). These results suggested that vasodilation for blood flow recovery after cerebral IR is more effective in aged rats and SHRs, which are at a risk of stroke, compared to young WRs with a normal blood pressure. This is thought to be the cause of decreased resilience and delayed blood flow recovery time.

We hypothesize that vasodilatory dysfunction and delayed blood flow recovery time after cerebral IR are caused by the cerebral endothelial vessel dysfunction associated with hypertension and aging.

The vascular endothelium releases endothelium-derived relaxing factor (EDRF) to relax the vascular smooth muscle [13], and it has been suggested that EDRF might be NO produced by eNOS. Furthermore, it has been reported in rats that EDRF dysfunction worsens with aging, and that it is even greater in SHRs [14]. Therefore, the decline in eNOS activity caused by hypertension and aging is thought to impair endothelial function, resulting in vasodilatory dysfunction and a delayed blood flow recovery time after cerebral ischemia, which is in line with the results of the current study.

However, a report suggested that eNOS protein and activity levels increase with age, but that the production of NO decreases [15]. Immunohistochemical studies are necessary to determine whether changes in eNOS-mediated NO production are affected by aging.

In our CBF data on the hippocampi of the SHRs, the rCBF was significantly lower in the older rats compared to the younger rats during ischemia. However, this was significantly higher in the older animals than in the younger animals after reperfusion, especially in the first 10 min after reperfusion.

The total NO data on the hippocampi of the SHRs did not differ significantly between the older and younger rats before or during ischemia, or after reperfusion.

We believe that vasodilatory dysfunction might be related to endothelium-associated changes in NO production, but that NO is not the only factor involved in the regulation of CBF. Other factors, including prostaglandins and endothelin-1, as well as neurotransmitters such as cathecholamine, might be related to the regulation of CBF.

In this experiment, we investigated the dynamics of NO production in the hippocampus and striatum. In the hippocampus, immunohistochemical studies showed that the nNOS mRNA expression was high [16,17], and differences from the striatum were reported [18], including an enhanced NO production during ischemia and hypoxia. Although the NO production in the WRs was not affected by aging, the effect of aging in the SHRs was clear. In both the hippocampus and striatum, the NO_3_^−^ levels were significantly lower in the older animals before ischemia and after IR, compared to the younger animals.

NO production requires L-arginine as a substrate. It is thought that, during cerebral ischemia, the supply of L-arginine decreases because of the blockage of cerebral blood flow, resulting in a decrease in NO production, and that reperfusion causes this supply to resume, resulting in an increase in NO production [16,19]. This increased NO production is thought to cause brain tissue damage by increasing free radicals during reperfusion after ischemia. However, there have been reports that it may contribute to resistance against cerebral ischemia [17,18] and that it exerts brain protective effects [19]. Moreover, it has been reported that L-arginine tends to decrease with aging [20], and that NOS also decreases [21].

In SHRs, aging and hypertension reduce their ability to synthesize NO in the brain, and as a result, the NO_3_^−^ levels after cerebral IR were lower in the older group than in the younger group. Furthermore, in our experiments, NO was collected from a probe inserted into the brain parenchyma. It was impossible to distinguish whether it was produced by nNOS or eNOS.

Some studies have indicated that neurons expressing nNOS (by immunostaining) are abundant in the cerebellum and olfactory system, and that only 2% of neurons in the cerebral cortex and hippocampus express this protein [22,23]. Therefore, the results obtained in this study may strongly reflect the changes in eNOS activity with aging.

Hypertension and aging reduce vascular eNOS activity and NO production, resulting in decreased vasodilation and a delayed blood flow recovery time after cerebral IR. These changes might account for the decline in NO_3_^−^ after cerebral IR in the aged rats and SHRs in the current study.

In our investigation of the kinetics of NO production in the brain, we quantitatively measured NO_2_^−^ and NO_3_^−^ using the Griess reaction, but it was not possible to distinguish between NO_2_^−^ and NO_3_^−^.

Yamada et al. investigated the NO in the cerebellum of rats, and found high levels of NO_3_^−^ after NMDA administration, noting that depolarization with potassium increased the NO_2_^−^ levels [22,24]. Cui et al. [24,25] and Doi et al. [23,25] reported that nNOS, a Ca^2+^-dependent enzyme, is activated by Ca^2+^ influx through NMDA receptors.

In our study, because the microdialysis probe was inserted near not only the capillaries, but also near neurons and astrocytes, it was very difficult to clarify the origin of the NO. In our study using knockout mice [3], we concluded that the NO production might reflect not only eNOS, but also nNOS.

This study suggested that aging causes a decrease in NO substrates, including L-arginine, and a decrease in NO levels. Whether this decline results from reduced eNOS or nNOS activity should be examined histopathologically using immunostaining with eNOS and nNOS antibodies.

## 5. Conclusions

Our in vivo findings suggested that, during aging, SHRs exhibit a reduction in their NO production, especially in the striatum, before and during cerebral ischemia, as well as after reperfusion. Hypertension and aging may be important factors affecting the NO production in a brain with IR injury.

## Figures and Tables

**Figure 1 ijms-24-12749-f001:**
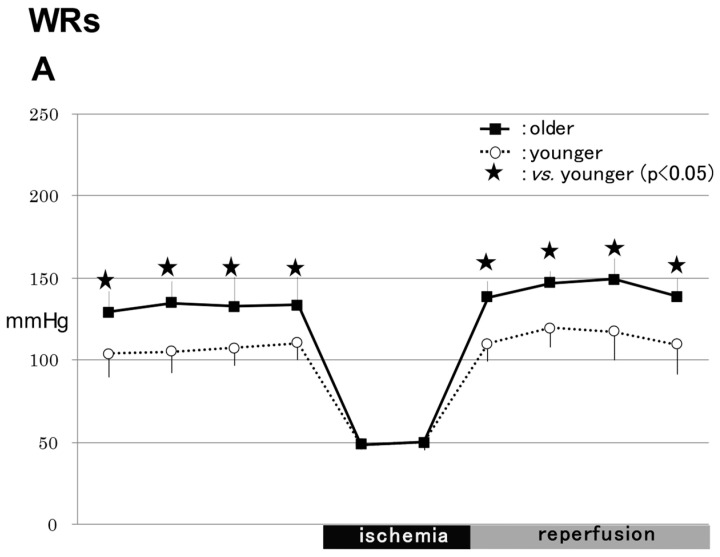
Mean arterial blood pressure. Changes in mean arterial blood pressure are shown in (**A**, **B**) The pre-ischemic mean arterial blood pressure (MABP) baselines are shown as averages of the last 40 min before ischemia. In the WRs, the MABPs were significantly higher in the older rats (145 ± 28 mmHg) than the younger rats (111 ± 27 mmHg) before ischemia and 10–40 min after reperfusion (*p* < 0.05) (**A**). In the SHRs, the MABPs were significantly higher in the older rats (166 ± 20 mmHg) than in the younger rats (118 ± 17 mmHg) 10 min after reperfusion (*p* < 0.05) (**B**). WRs, Wistar rats; SHRs, spontaneously hypertensive rats.

**Figure 2 ijms-24-12749-f002:**
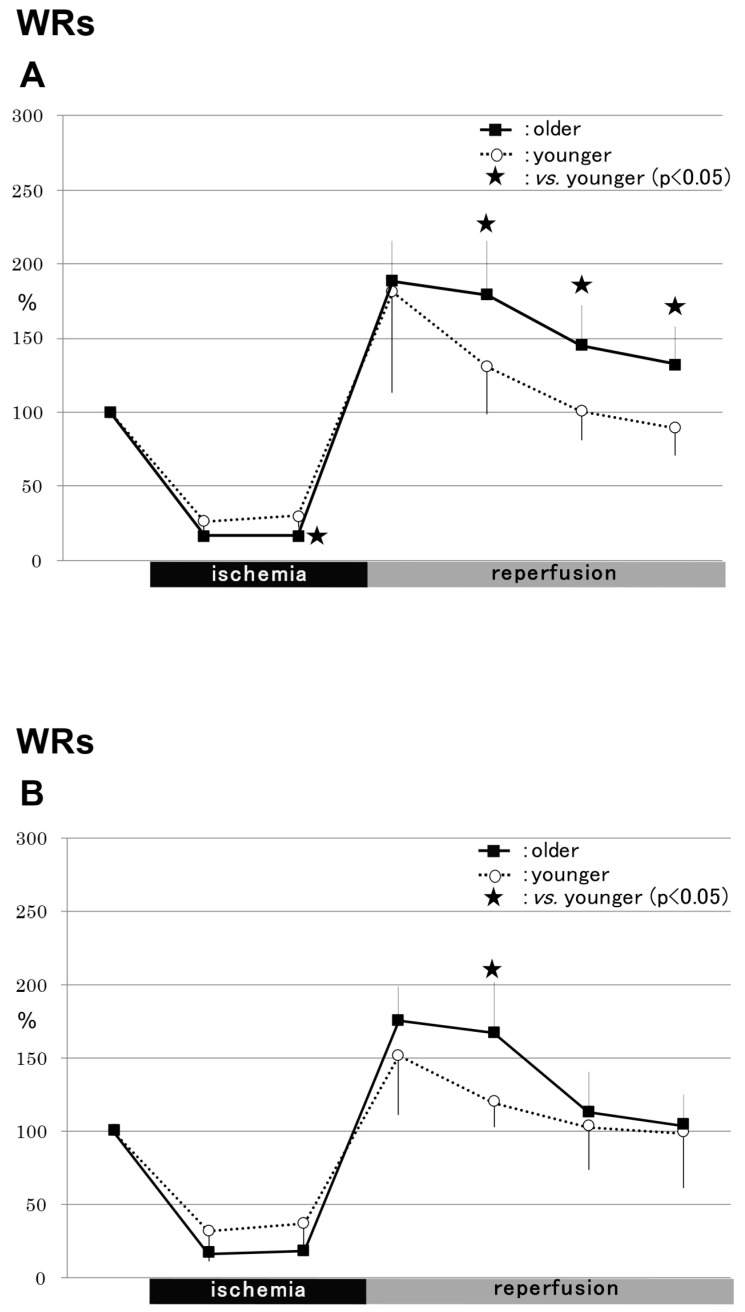
Regional cerebral blood flow. Changes in regional cerebral blood flow are shown in (**A**–**D**). In the striatums in the WR group, the rCBF was significantly lower in the older rats (16.4 ± 10.4% of baseline) compared with the younger rats (30.2 ± 7.5%) during ischemia (*p* < 0.05) (**A**). In contrast, the rCBF was significantly higher in the older rats (179.3 ± 35.8% of baseline) than in the younger rats (130.7 ± 31.3%) 20–40 min after reperfusion (*p* < 0.05) (**A**). In the WR group, the hippocampal rCBF was significantly higher in the older rats (166.9 ± 35.7% of baseline) than in the younger rats (119.5 ± 16.3%) 20 min after reperfusion (*p* < 0.05) (**B**). In the striatums of the SHR group, the rCBF values were similar between the two age groups (**C**). In contrast, in the hippocampi of the SHR group, the rCBF was significantly lower in the older rats (17.9 ± 6.3% of baseline) compared to the younger rats (35.8 ± 12.4%) during ischemia (*p* < 0.05), and significantly higher in the older animals (213.6 ± 48.7% of baseline) than in the younger animals (163.8 ± 33.6%) 10 min after reperfusion (*p* < 0.05) (**D**). WRs, Wistar rats; SHRs, spontaneously hypertensive rats.

**Figure 5 ijms-24-12749-f005:**
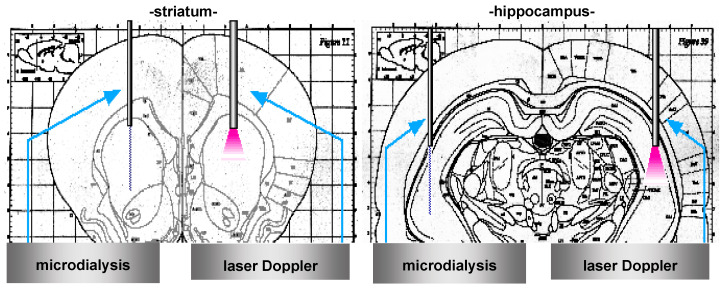
Probe set up. Laser Doppler probes (Advance Laser Flowmeter ALF21D; Advance, Tokyo, Japan) were inserted into the right striatum and hippocampus to continuously measure the regional cerebral blood flow. The production of NO was continuously monitored using in vivo microdialysis. A microdialysis probe was inserted into the left striatum and hippocampus and perfused with Ringer’s solution at a constant rate of 2 µL/min. After a 2 h equilibration period, fractions were collected every 10 min using a micro-fraction collector (EF-80B; Eicom, Kyoto, Japan).

## Data Availability

The data presented in this study are available upon request from the corresponding author.

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
