# Peer review of "The Effect of Aging on Nitric Oxide Production during Cerebral Ischemia and Reperfusion in Wistar Rats and Spontaneous Hypertensive Rats: An In Vivo Microdialysis Study"

_ijms, 2023, doi:10.3390/ijms241612749_

Round 1

Reviewer 1 Report

Some critical concerns need to be improved or answered:

1. The authors only described what are their research findings in previous studies (To some extend, too many self-citation?) in INTRODUCTION. The rationale of this study, i.e. why the influence of hypertension and aging on NO production during cerebral ischemia and reperfusion is worthy of investigation, should be described clearly in a logical way.

2. What is the dosage of pentobarbital sodium used in this study?

 3. A polyethylene catheter with the size of PE-10 was inserted into rats' femoral artery to moniter their blood pressure. However, a PE-50 catheter is empirically suitable for monitering rat's blood pressure via femoral artery insertion. The diameter of PE-10 is too small to be fixed on femoral artery and to transmit blood pressure signal. The blood pressure will be stunted by the small diameter.

4. The interpretations of the results are too rough.

5. In Wistar rats, the results showed that aging significantly increases MABP. Why similar result could not be found in the SHR rats?

6. The authors hypothesize that vasodilatory dysfunction might be related to endothelium-associated change of NO production. However, why was the rCBF decreased in hippocampus of SHR without significant change in NO production in the hippocampus of SHR?

7. With a microdialysis analysis, how can the authors differentiate the origins of NO produced? Is it coming from eNOS or nNOS?

8. A legend is required for all figures.

Reviewer 2 Report

Ito et al. reported effects of aging on nitric oxide production in the model of SHR, comparing 12-month-old animals with 3-month-old animals. Their conclusions are that older SHR rats had less NO production in the brain. This is an interesting study with potentially valuable data presented. However, there are several issues, which should be clarified before publication. 

Major issues: 

1. Generally, the lifespan of rats are 2.5-3 years. The ages used in the current study was 3-month-old and 12-month-old, and the authors claimed 12-month-old animals as "older". Please add discussion, that how "old" are these animals, and what are the approximate equivalent life stage in human? 

2. All the comparisons were made within WR or SHR animals. Could you please also compare cross board? It would be interesting to see how NO exerts biological/pathological roles in the hypertension/reperfusion damage. 

3. As-is, the current study did not report the infarct volume of the brain, which is routine method to evaluate the severity of ischemic and reperfusion damage. Please add the data or explain why they were absent. 

Minor issues:

1. Why the sample sizes in older animals were smaller than the younger animals? 

2. The language needs further proof-reading.

3. Some of the figures seems to be cropped (incomplete title on the top-left corner). Figure legends seemed to be missing for all the figures. 

Ito et al. reported effects of aging on nitric oxide production in the model of SHR, comparing 12-month-old animals with 3-month-old animals. Their conclusions are that older SHR rats had less NO production in the brain. This is an interesting study with potentially valuable data presented. However, there are several issues, which should be clarified before publication. 

Major issues: 

1. Generally, the lifespan of rats are 2.5-3 years. The ages used in the current study was 3-month-old and 12-month-old, and the authors claimed 12-month-old animals as "older". Please add discussion, that how "old" are these animals, and what are the approximate equivalent life stage in human? 

2. All the comparisons were made within WR or SHR animals. Could you please also compare cross board? It would be interesting to see how NO exerts biological/pathological roles in the hypertension/reperfusion damage. 

3. As-is, the current study did not report the infarct volume of the brain, which is routine method to evaluate the severity of ischemic and reperfusion damage. Please add the data or explain why they were absent. 

Minor issues:

1. Why the sample sizes in older animals were smaller than the younger animals? 

2. The language needs further proof-reading.

3. Some of the figures seems to be cropped (incomplete title on the top-left corner). Figure legends seemed to be missing for all the figures. 

Round 2

Reviewer 2 Report

The authors have addressed most of my concerns and the manuscript should be acceptable. However, I recommend them to include more details in the figure legends, so the figures may be easier interpreted. 
